# Effect of inspiratory muscle training in esophageal cancer patients receiving esophagectomy: A meta-analysis of randomized controlled trials

**Jianhua Su, Wei Huang, Pengming Yu** *

Rehabilitation Medicine Center, West China Hospital, Sichuan University, Chengdu, Sichuan Province, China

* Homer.yu@wchscu.edu.cn.

## Abstract

### Purpose

To identify the clinical effect of inspiratory muscle training (IMT) among esophageal cancer patients undergoing esophagectomy based on randomized controlled trials (RCTs).

### Methods

Several databases were searched for relevant RCTs up to August 23, 2023. Primary outcomes were respiratory muscle function, including the maximum inspiratory pressure (MIP) and maximum expiratory pressure (MEP), and pulmonary function, including the forced expiratory volume in one second % (FEV1%), forced vital capacity% (FVC%), maximal ventilator volume (MVV), FEV1/FVC% and FVC. The secondary outcomes were exercise performance, including the six-minute walk distance test (6MWT) and Borg index; mental function and quality of life, as evaluated by the Hospital Anxiety Depression Scale (HADS) and Nottingham Health Profile (NHP) score; and postoperative complications. All the statistical analyses were performed with REVMAN 5.3 software.

### Results

Eight RCTs were included in this meta-analysis, with 368 patients receiving IMT and 371 control subjects. The pooled results demonstrated that IMT could significantly enhance respiratory muscle function (MIP: MD = 7.14 cmH2O, P = 0.006; MEP: MD = 8.15 cmH2O, P<0.001) and pulmonary function (FEV1%: MD = 6.15%, P<0.001; FVC%: MD = 4.65%, P<0.001; MVV: MD = 8.66 L, P<0.001; FEV1/FVC%: MD = 5.27%, P = 0.03; FVC: MD = 0.50 L, P<0.001). Furthermore, IMT improved exercise performance (6MWT: MD = 66.99 m, P = 0.02; Borg index: MD = -1.09, P<0.001), mental function and quality of life (HADS anxiety score: MD = -2.26, P<0.001; HADS depression score: MD = -1.34, P<0.001; NHP total score: MD = -48.76, P<0.001). However, IMT did not significantly decrease the incidence of postoperative complications.

**Data Availability Statement:** All data generated or analyzed during this study are included in this published article.

**Funding:** The author(s) received no specific funding for this work.

**Competing interests:** The authors have declared that no competing interests exist.

## Conclusion

IMT improves clinical outcomes, such as respiratory muscle function and pulmonary function, in esophageal cancer patients receiving esophagectomy and has potential for broad applications in the clinic.

## Introduction

Esophagectomy is an important clinical treatment for esophageal cancer patients. Despite the increasing prevalence of minimally invasive approaches in recent decades, esophageal cancer surgery continues to cause significant trauma. Additionally, the use of anesthesia, sedation, analgesics, and damage to the diaphragm and chest wall muscles can lead to a substantial decline in postoperative cardiopulmonary function and physical mobility of patients [1–5]. The incidence of complications after esophageal cancer surgery remains relatively high [6]. Maintaining good respiratory status is of paramount importance for promoting rapid recovery in postoperative esophageal cancer patients. These issues can also impose a significant psychological burden on patients, seriously affecting their quality of life [7].

The conventional respiratory training methods currently used in clinical practice commonly include diaphragmatic breathing, blowing up balloons, and lip puckering breathing. These methods have various advantages, such as simplicity, convenience, and cost-effectiveness; they are also easy for patients to accept and, with regular training, can effectively improve the respiratory function of patients [8, 9]. However, conventional respiratory training has limitations due to factors such as the monotony of training, difficulty in controlling the optimal intensity of respiratory training, and inaccurate mastery of respiratory movements, which restrict training effectiveness.

Previous studies have demonstrated the clinical value of inspiratory muscle training (IMT) in pulmonary, cardiac and abdominal surgeries, as it effectively improves patients' respiratory muscle function and postoperative outcomes [10–13]. IMT can enhance inspiratory muscle strength, alleviate inspiratory muscle tension, improve diaphragm function and promote lung expansion, thereby assisting in maintaining airway patency [13, 14]. Furthermore, IMT can play a role in inhibiting sympathetic nervous system function, enhancing vagus nerve activity, and reducing peripheral vascular resistance [15–17].

Esophageal cancer surgery may involve the removal of part of the esophagus or adjacent organs, affecting the tissues within the chest cavity; this can impact the patient's respiratory function, including lung capacity and respiratory muscle strength. IMT can help patients regain respiratory function and increase lung capacity and respiratory muscle strength [13, 14]. Following esophageal cancer surgery, patients may experience respiratory complications such as lung infections, pneumonia, or atelectasis. These complications can prolong recovery time and increase patient discomfort. Inspiratory muscle training can help prevent the occurrence of these respiratory complications, improving the success rate of recovery [15]. Esophageal cancer surgery is a traumatic procedure that can lead to physical weakness and muscle atrophy in patients. IMT can assist patients in increasing muscle strength and endurance, accelerating the recovery process and reducing postoperative complications [16]. In addition, surgery is a crucial treatment method for esophageal cancer, but it may also have certain impacts on patients' lives, such as restricted activity, pain, and discomfort. IMT enables patients to enhance their physical fitness, improve their quality of life, and better adapt to life after surgery. Therefore, theoretically, IMT may also have significant clinical value for

esophageal cancer surgery patients, but currently, there is a lack of robust strong evidence supporting this topic.

Some researchers have suggested that applying inspiratory muscle training in esophageal cancer surgery may also yield favorable clinical results. Several relevant randomized controlled trials (RCTs) have explored the clinical effect of IMT in esophageal cancer patients receiving esophagectomy, but their results have shown significant differences regarding various factors, such as the impact of IMT on respiratory muscle function and pulmonary function.

Thus, the aim of this meta-analysis was to further identify the effects of IMT on clinical outcomes in esophageal cancer patients receiving esophagectomy based on currently available RCTs.

## Materials and methods

The current meta-analysis was performed according to the Preferred Reporting Items for Systematic Review and Meta-Analyses 2020 [18].

### Ethical statement

The authors are accountable for all aspects of the work in ensuring that questions related to the accuracy or integrity of any part of the work are appropriately investigated and resolved. All procedures performed in studies that involved human participants were in accordance with the ethical standards of the institutional and/or national research committee and with the 1964 Helsinki Declaration and its later amendments or comparable ethical standards.

### Literature search

The Medline EMBASE, Web of Science, Cochrane Library and China National Knowledge Infrastructure (CNKI) databases were searched from their inception to August 23, 2023. The following terms were used during the search: inspiratory muscle training, IMT, esophageal, esophagus, tumor, neoplasm, cancer and carcinoma. The detailed search strategy was as follows: (inspiratory muscle training OR IMT) AND (esophageal OR esophagus) AND (tumor OR neoplasm OR cancer OR carcinoma). MeSH terms and free texts were applied, and references cited in the included studies were also reviewed.

### Inclusion criteria

In our meta-analysis, the following inclusion criteria were used: 1) patients who underwent esophagectomy and were diagnosed with primary esophageal cancer; 2) patients who were RCTs; 3) patients in the experimental group who received IMT through a respiratory training device before or (and) after surgery for at least one week and patients in the control group who did not receive IMT; 4) patients whose clinical outcomes are described below; and 5) patients whose clinical outcomes were published in English or Chinese.

### Exclusion criteria

Studies that met the following criteria were excluded: 1) had overlapping or duplicated data; 2) had insufficient data for the calculation of clinical outcomes; 3) had other interventions combined; 4) had meeting abstracts, letters, animal trials, case reports or reviews; and 5) had low-quality studies with a PEDro score of 3 or lower [19, 20].

### Data collection

The following information was extracted from each included study: the name of the first author, publication year, country, number of participants, type of surgery, type of IMT, age,

intervention time, initial training pressure, training time, sessions and duration of IMT, control care, endpoints, information about the PEDro scale and detailed data about observation endpoints, including the mean differences (MDs) and odds ratios (ORs) with 95% confidence intervals (CIs).

The primary outcomes were respiratory muscle function, including the maximum inspiratory pressure (MIP) and maximum expiratory pressure (MEP), and pulmonary function, including the forced expiratory volume in one second % (FEV1%), forced vital capacity% (FVC%), maximal ventilator volume (MVV), FEV1/FVC% and FVC. The secondary outcomes were exercise performance, including the six-minute walk distance test (6MWT) and Borg index; mental function and quality of life, assessed by the Hospital Anxiety Depression Scale (HADS) and Nottingham Health Profile (NHP) score; and postoperative complications, assessed by pulmonary postoperative complications, pulmonary infection, anastomotic fistula, retube, chylothorax, vocal cord paralysis, atelectasis, pleural effusion, wound infection and cardiac complications.

### Methodological quality assessment

In this meta-analysis, the methodological quality was assessed according to the PEDro scale. Studies with a PEDro score of 6 or higher, 4 or 5 and 3 or lower were defined as high-, fair- and low-quality studies, respectively [19, 20].

The literature search, selection, data collection and quality assessment were all conducted by two investigators, and any disagreements were resolved by team discussion.

### Statistical analysis

All the statistical analyses were conducted with RevMan version 5.3 software. The heterogeneity between studies was quantified by the $I^2$ statistic and Q test. If significant heterogeneity was observed, represented by $I^2 > 50\%$ and/or $P < 0.10$, the random effects model was applied; otherwise, the fixed effects model was used [21, 22]. Continuous data were compared and analyzed as the changes from baseline values at one of the following time points: at admission, before the intervention or operation to final values at one of the following time points: at discharge, after the intervention or an interval after the surgery. The mean differences (MDs) with standard deviations (SDs) were combined to calculate the MDs with corresponding 95% CIs between the experimental and control groups. The data are presented as the means and ranges and were converted to means and SDs according to the formula reported by Hozo et al. [23]. Discontinuous data were compared and then represented with ORs and 95% CIs. A P value < 0.05 was considered to indicate statistical significance.

## Results

### Literature search and selection

As shown in **Fig 1**, 77 records were initially identified from several databases, and 25 duplicate records were removed. After reviewing the titles and abstracts, 41 publications were excluded. After reviewing the full texts and excluding three studies, eight RCTs were eventually included in this meta-analysis [24–31].

### Basic characteristics of the included studies

A total of 739 patients were enrolled, including 368 and 371 patients who received and did not receive IMT, respectively. These studies were published between 2014 and 2023, and the sample sizes ranged from 39 to 241. Most of the included studies were from China. Preoperative

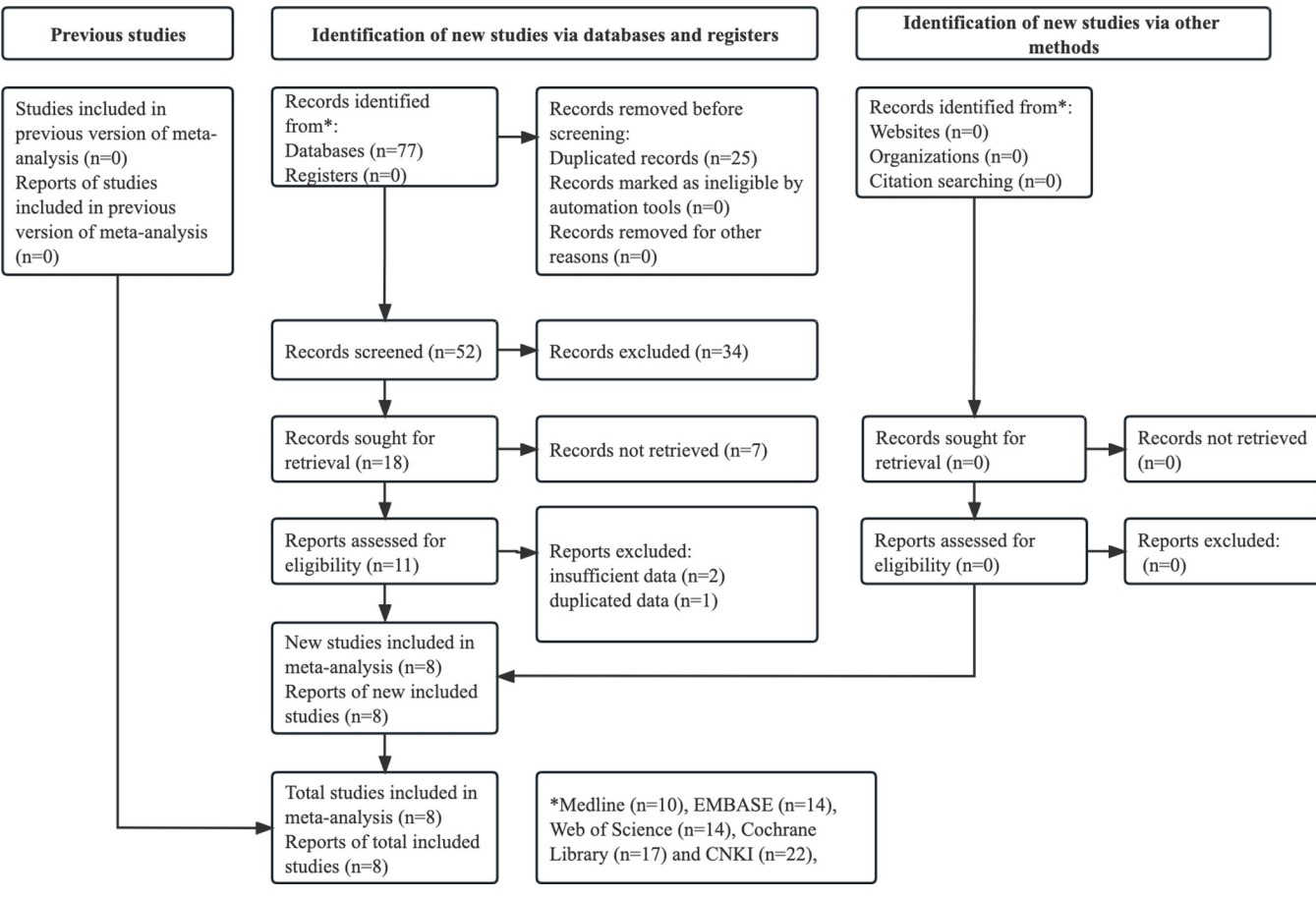

**Fig 1. Prisma flow diagram of this meta-analysis.**

IMT was involved in most patients. In addition, 60% of the initial training pressure and seven sessions a week were applied in most of the included studies. The durations of IMT ranged from 10 days to 4 weeks. Other specific information, including the parameters of the IMT, is clearly presented in **Table 1**.

## Methodological quality

Detailed information about the PEDro score of each included study is shown in **Table 2**. All included studies were of high quality, with a PEDro score of 5 or higher.

## Primary outcomes

**Respiratory muscle function.** Five and four studies explored the effect of perioperative IMT on the MIP and MEP, respectively, in patients undergoing esophagectomy. The pooled results demonstrated that IMT significantly increased the MIP (MD = 7.14%, 95% CI: 2.05–12.20%, P = 0.006; $I^2$ = 87%, P<0.001) (**Fig 2A**) and MEP (MD = 8.15%, 95% CI: 4.76–11.54%, P<0.001; $I^2$ = 72%, P = 0.01) (**Fig 2B**) (**Table 3**).

**Pulmonary function.** Similarly, IMT plays a role in improving pulmonary function in surgical esophageal cancer patients. Patients in the IMT group had increased FEV1% (MD = 6.15%, 95% CI: 4.06–8.23%, P<0.001; $I^2$ = 72%, P = 0.01) (**Fig 3A**), FVC% (MD = 4.65%, 95% CI: 2.70–6.60%, P<0.001; $I^2$ = 0%, P>0.999) (**Fig 3B**), MVV (MD = 8.66 L,

**Table 1. Basic characteristics of included studies.**

| Author | Year | Country | Number (T/C) | Surgery type | Age (year-old) | Type of IMT | Intervention time | Initial training pressure (%) | Training time (min/day) | Sessions (n/week) | Duration | Control | Endpoints |
|---|---|---|---|---|---|---|---|---|---|---|---|---|---|
| Van Adrichem [24] | 2014 | Netherlands | 20/19 | Open | 62.0 ±7.1 | Breath training device | Preoperative | 60 | 6 times | 3 | 3 weeks | Regular care | ④ |
| Wei [25] | 2015 | China | 30/30 | NR | 50–75 | Breath training device | Preoperative and Postoperative | NR | 20 minutes*4 times | 7 | 10 days | Regular care | ②④⑤ |
| Huang [26] | 2017 | China | 45/45 | NR | 50–75 | Breath training device | Preoperative and Postoperative | NR | 20 minutes*4 times | 7 | 10 days | Regular care | ①②③④⑤ |
| Guinan [27] | 2018 | Multiple countries | 28/32 | Open or Thoracoscope | 64.13 ±7.8 | Breath training device | Preoperative | 60 | 30 times*2 | 7 | 2 weeks | Regular care | ①③④ |
| Valkenet [28] | 2018 | Multiple countries | 120/121 | Open or Thoracoscope | 63.1 ±7.5/ 62.7 ±8.9 | Breath training device | Preoperative | 60 | 30 times*2 | 7 | 2 weeks | Regular care | ④ |
| Deng [29] | 2020 | China | 40/40 | Open or Thoracoscope | 50–78 | Breath training device | Preoperative and Postoperative | NR | 15∼20 minutes*4 times | 7 | 10 days | Regular care | ①③⑤ |
| Hu [30] | 2021 | China | 40/40 | Open or Thoracoscope | 48–72 | Breath training device | Preoperative and Postoperative | NR | 15∼20 minutes*4 times | 7 | 10 days | Regular care | ①②③ |
| Gao [31] | 2023 | China | 45/44 | NR | 59.86 ±7.26 | Breath training device | Postoperative | NR | 30 times*2 | 5 | 4 weeks | Regular care | ①②③④ |

T: test group; C: control group; NR: not reported; ① respiratory muscle function; ② respiratory muscle function; ③ exercise performance; ④ postoperative complication; ⑤ Mental function and quality of life; IMT: inspiratory muscle training.

95% CI: 7.17–10.14 L, P<0.001; $I^2$ = 17%, P = 0.30) (**Fig 3C**), FEV1/FVC% (MD = 5.27%, 95% CI: 0.39–10.16%, P = 0.03; $I^2$ = 87%, P = 0.005) (**Fig 3D**) and FVC (MD = 0.50 L, 95% CI: 0.32–0.68 L, P<0.001). (**Table 3**)

**Table 2. Quality assessment for included trials according to PEDro sale.**

| Study | 1 | 2 | 3 | 4 | 5 | 6 | 7 | 8 | 9 | 11 | Overall score |
|---|---|---|---|---|---|---|---|---|---|---|---|
| Van Adrichem [24] | Y | Y | Y | Y | N | N | Y | Y | Y | Y | 8/10 |
| Wei [25] | Y | Y | N | Y | N | N | N | Y | Y | Y | 5/10 |
| Huang [26] | Y | Y | N | Y | N | N | N | Y | Y | Y | 5/10 |
| Guinan [27] | Y | Y | Y | Y | N | N | N | Y | Y | Y | 7/10 |
| Valkenet [28] | Y | Y | Y | Y | N | N | N | Y | Y | Y | 7/10 |
| Deng [29] | Y | Y | N | Y | N | N | N | Y | Y | Y | 5/10 |
| Hu [30] | Y | Y | N | Y | N | N | N | Y | Y | Y | 5/10 |
| Gao [31] | Y | Y | N | Y | N | N | N | Y | Y | Y | 5/10 |

N: no criteria or not satisfied; Y: yes (criteria satisfied).

1: eligibility criteria, 2: random allocation, 3: concealed allocation, 4: baseline comparability, 5: blind subjects, 6: blind therapists, 7: blind assessors, 8: adequate follow-up, 9: intention-to-treat analysis, 10: between-group comparisons 11: point estimates and variability.

The total PEDro score is the sum of items 2 to 11, which relate to internal validity. Item 1 is reported to indicate external validity.

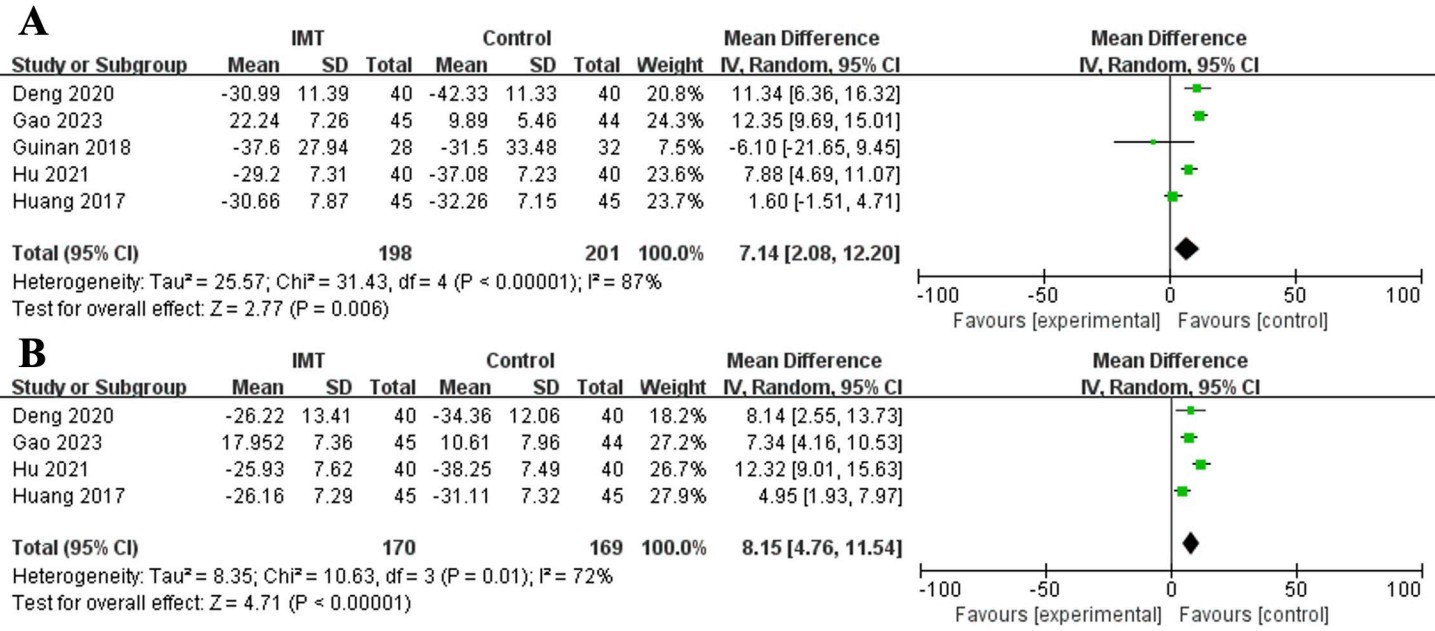

**Fig 2.** Forest plots for the effect of inspiratory muscle training on maximal inspiratory pressure (A) and maximum expiratory pressure (B).

### Secondary outcomes

**Exercise performance.** Based on previous research, the 6MWT and Borg scale are widely utilized for assessing exercise performance status among patients receiving rehabilitation training, demonstrating good applicability and reliability. Therefore, in this meta-analysis, the 6MWT and Borg index were used to assess exercise performance status. The pooled results indicated that IMT could significantly increase the 6MWT (MD = 66.99 m, 95% CI: 10.13–123.85 m, P = 0.02; $I^2$ = 93%, P<0.001) (**Fig 4A**) and decrease the Borg index (MD = -1.09, 95% CI: -1.31 ∼ -0.88, P<0.001; $I^2$ = 81%, P = 0.001) (**Fig 4B**) (**Table 3**).

**Mental function and quality of life.** The HADS was used to evaluate mental function, and pooled results revealed that patients in the IMT group had significantly lower HADS anxiety scores (MD = -2.26, 95% CI: -3.42 ∼ -1.10, P<0.001; $I^2$ = 64%, P = 0.06) (**Fig 5A**) and HADS depression scores (MD = -1.34, 95% CI: -1.89 ∼ -0.79, P<0.001; $I^2$ = 0%, P = 0.66) (**Fig 5B**). In addition, the NHP total score was used to assess quality of life, and our results indicated that IMT obviously improved quality of life in patients who underwent esophagectomy (MD = -48.76, 95% CI: -55.24 ∼ -42.28, P<0.001; $I^2$ = 0%, P = 0.67) (**Fig 5C**). (**Table 3**).

**Postoperative complications.** We assessed the effect of IMT on postoperative complications based on available data provided in the included studies. Overall, IMT did not decrease the risk of postoperative complications, including postoperative pulmonary complications (OR = 0.57, P = 0.08; $I^2$ = 14%, P = 0.32), pulmonary infections (OR = 0.88, P = 0.57; $I^2$ = 20%, P = 0.29), anastomotic fistulas (OR = 1.08, P = 0.82; $I^2$ = 0%, P = 0.97), vomiting (OR = 0.43, P = 0.45; $I^2$ = 57%, P = 0.13), chylothorax (OR = 1.54, P = 0.28; $I^2$ = 37%, P = 0.21), vocal cord paralysis (OR = 0.35, P = 0.06; $I^2$ = 0%, P = 0.73), atelectasis (OR = 0.52, P = 0.34; $I^2$ = 0%, P = 0.66), pleural effusion (OR = 1.00, P>0.999; $I^2$ = 0%, P>0.999), or wound infection (OR = 1.49, P = 0.41; $I^2$ = 0%, P = 0.72). Detailed information is presented in **Table 3**.

**Table 3. Results of meta-analysis.**

| | Number of studies | MD/OR | 95% CI | P value | $I^2$ (%) | P value |
|---|---|---|---|---|---|---|
| **Primary outcomes** | | | | | | |
| Respiratory muscle function | | | | | | |
| MIP (cmH$_2$O) | 5 | 7.14 | 2.08–12.20 | **0.006** | 87 | <0.001 |
| MEP (cmH$_2$O) | 4 | 8.15 | 4.76–11.54 | **<0.001** | 72 | 0.01 |
| Pulmonary function | | | | | | |
| FEV1% | 4 | 6.15 | 4.06–8.23 | **<0.001** | 72 | 0.01 |
| FVC% | 2 | 4.65 | 2.70–6.60 | **<0.001** | 0 | >0.999 |
| MVV (L) | 3 | 8.66 | 7.17–10.14 | **<0.001** | 17 | 0.30 |
| FEV1/FVC% | 2 | 5.27 | 0.39–10.16 | **0.03** | 87 | 0.005 |
| FVC (L) | 1 | 0.50 | 0.32–0.68 | **<0.001** | - | - |
| **Secondary outcomes** | | | | | | |
| Exercise performance | | | | | | |
| 6MWT (m) | 4 | 66.99 | 10.13–123.85 | **0.02** | 93 | <0.001 |
| Borg index | 4 | -1.09 | -1.31–0.88 | **<0.001** | 81 | 0.001 |
| Mental function and quality of life | | | | | | |
| HADS anxiety score | 3 | -2.26 | -3.42–1.10 | **<0.001** | 64 | 0.06 |
| HADS depression score | 3 | -1.34 | -1.89–0.79 | **<0.001** | 0 | 0.66 |
| NHP total score | 3 | -48.76 | -55.24–42.28 | **<0.001** | 0 | 0.67 |
| Postoperative complication | | | | | | |
| Pulmonary postoperative complication | 4 | 0.57 | 0.30–1.07 | 0.08 | 14 | 0.32 |
| Pulmonary infection | 5 | 0.88 | 0.56–1.38 | 0.57 | 20 | 0.29 |
| Anastomotic fistula | 2 | 1.08 | 0.54–2.17 | 0.82 | 0 | 0.97 |
| Retube | 2 | 0.43 | 0.05–3.78 | 0.45 | 57 | 0.13 |
| Chylothorax | 2 | 1.54 | 0.70–3.38 | 0.28 | 37 | 0.21 |
| Vocal cords paralysis | 2 | 0.35 | 0.12–1.05 | 0.06 | 0 | 0.73 |
| Atelectasis | 3 | 0.52 | 0.14–1.96 | 0.34 | 0 | 0.66 |
| Pleural effusion | 2 | 1.00 | 0.14–7.30 | >0.999 | 0 | 0.999 |
| Wound infection | 2 | 1.49 | 0.58–3.84 | 0.41 | 0 | 0.72 |
| Cardiac complication | 2 | 0.77 | 0.44–1.38 | 0.38 | 0 | 0.57 |

MIP: maximum inspiratory pressure; MEP: maximum expiratory pressure; FEV1: forced expiratory volume in one second; FVC: forced vital capacity; MVV: maximal ventilator volume; 6MWT: six-minute walk distance test; HADS: hospital anxiety depression scale; NHP: Nottingham Health Profile.

## Discussion

In this meta-analysis, we explored the clinical effects of IMT among esophageal cancer patients undergoing esophagectomy based on currently available RCTs. According to our pooled results, IMT has a role in enhancing respiratory muscle function and pulmonary function and improving exercise performance, mental function and quality of life in esophageal cancer patients receiving esophagectomy. However, no significantly decreased risk of postoperative complications was observed in the IMT group. Therefore, perioperative IMT could significantly improve clinical outcomes, such as respiratory muscle function and pulmonary function, in esophageal cancer patients and facilitate patient recovery after surgery, which shows high clinical application potential.

Several meta-analyses have clarified the clinical value of IMT in patients undergoing major surgery. Cordeiro et al. included seven RCTs and demonstrated that postoperative IMT significantly improved MIP (MD = 15.77 cmH2O, 95% CI: 5.95–25.49 cmH2O), MEP (MD = 15.87 cmH2O, 95% CI: 1.16–30.58 cmH2O), peak expiratory flow (PEF) (MD = 40.98 L/min, 95%

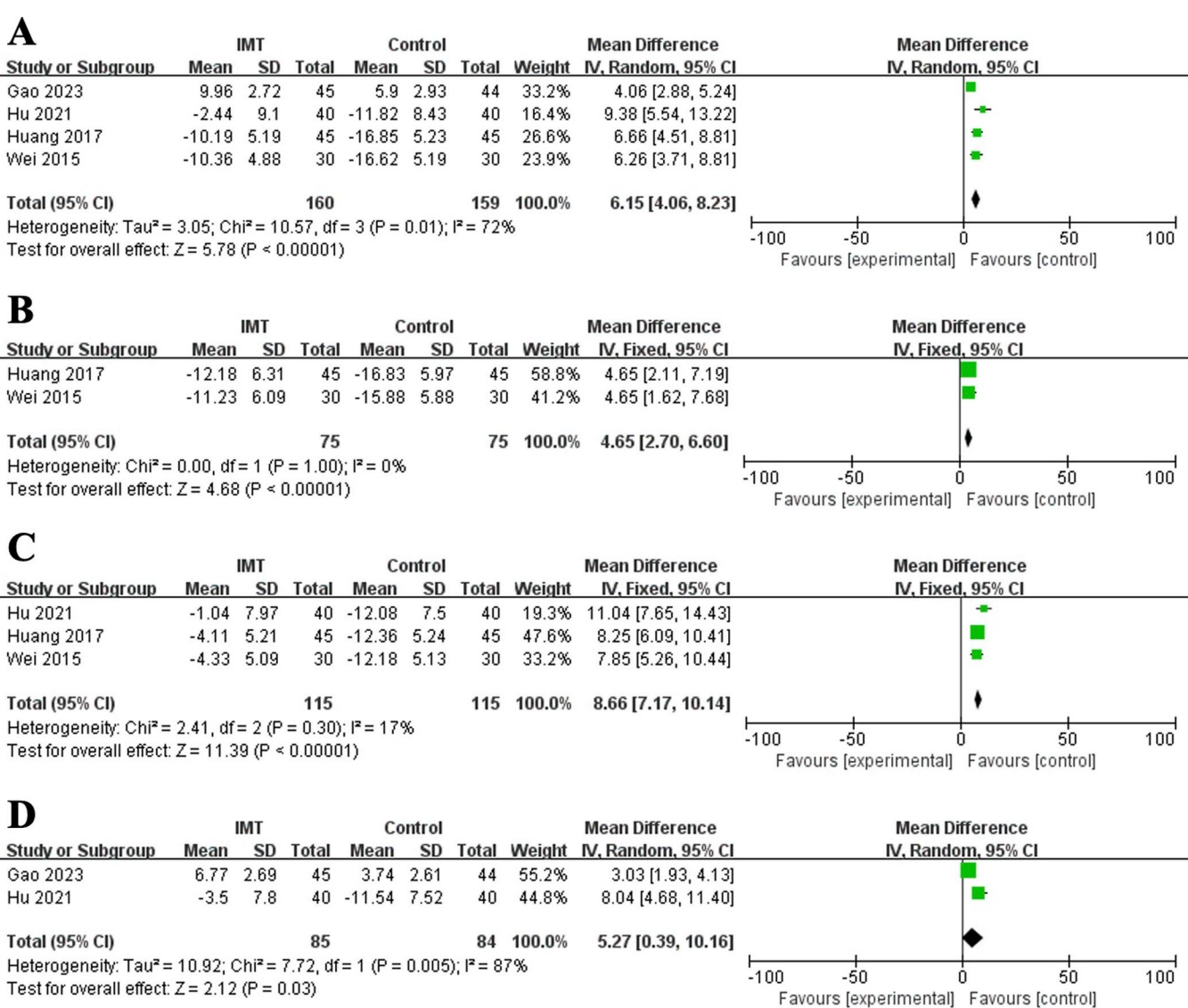

**Fig 3.** Forest plots for the effect of inspiratory muscle training on forced expiratory volume in one second% (A), forced vital capacity% (B), maximal ventilator volume (C) and forced expiratory volume in one second/forced vital capacity% (D).

CI: 4.64–77.32 L/min) and tidal volume (TV) (MD = 184.75 mL, 95% CI: 19.72–349.77 mL) and decreased hospital stay (MD = -1.25 days, 95% CI: -1.77 ∼ 0.72 days) in patients after myocardial revascularization [32]. Similarly, a meta-analysis by Katsura et al. indicated that preoperative IMT played a role in reducing the incidence of postoperative pulmonary complications [risk ratio (RR) = 0.53, 95% CI: 0.34–0.82] and decreasing the hospital stay (MD = -1.33 days, 95% CI: -2.53 ∼ 0.13 days) in patients undergoing cardiac and major abdominal surgery, such as myocardial revascularization, colon surgery and upper abdominal surgery, after 12 RCTs including 695 participants were included [10]. Furthermore, Yang et al. included seven studies and reported that perioperative IMT obviously improved the MIP (MD = 9.53 cmH2O, 95% CI: 3.98–15.08 cmH2O) [10]. Thus, the IMT may also have significant clinical value for improving postoperative respiratory function and reducing the risk of postoperative complications among patients who undergo esophageal cancer surgery. However, these meta-analyses

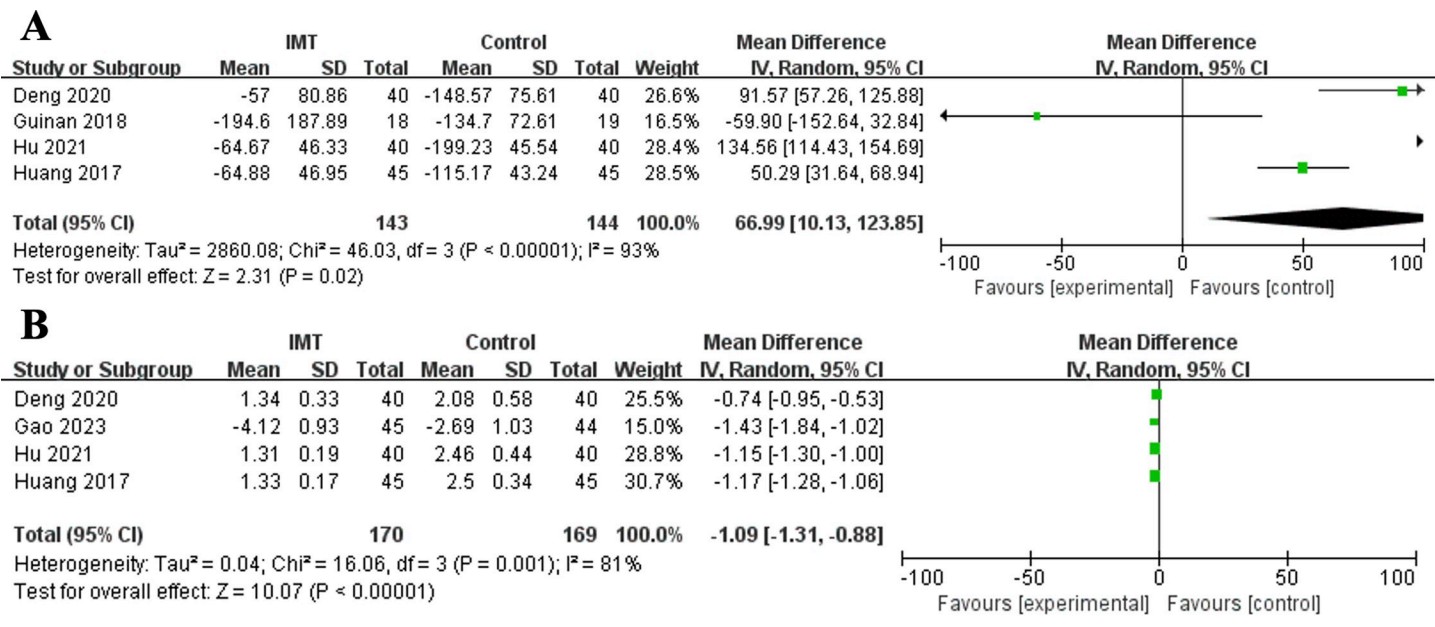

**Fig 4.** Forest plots for the effect of inspiratory muscle training on six-minute walk distance test (A) and Borg index (B).

included patients who underwent different surgical procedures and had various outcome measures and thus failed to provide conclusive evidence on whether esophageal cancer patients should receive IMT. In our study, we assessed the ability of IMT to improve postoperative outcomes among patients who underwent esophagectomy.

Notably, in five domestic studies, preoperative and postoperative training regimens were applied, and they yielded relatively consistent positive results in terms of improving patients' respiratory muscle function, lung function, exercise capacity, psychological function, and quality of life. However, patients in the three foreign studies only received preoperative training for 2 to 3 weeks, and the study by Guinan et al. showed no significant value of IMT in improving postoperative MIP or the 6MWT in esophageal cancer patients [27]. The authors speculate that there may be several reasons for this phenomenon. First, compared to preoperative training alone, the combination of preoperative and postoperative training may yield better results, but this needs further confirmation by more high-quality RCTs. Second, the clinical outcome measures analyzed in some foreign studies were generally limited. In detail, the studies by Van Adrichem and Valkenet et al. only explored the correlation between preoperative IMT and postoperative complications [24, 28]. Therefore, these data may not sufficiently demonstrate the clinical application value of IMT. Third, there may be significant differences between domestic and foreign studies in terms of patient populations, medical conditions, and other factors, leading to greater heterogeneity in the results. For example, the pathological types of esophageal cancer and the prevalence of minimally invasive surgery vary among different countries. Thus, our findings are to some extent primarily applicable to the domestic population.

Additionally, we believe that there are still some valuable fields about the clinical effects of IMT for esophageal cancer surgery that are worthy of further investigation. As mentioned above, the optimal timing for IMT intervention is worth further exploration. The optimal training regimen, including training time and duration, for different populations also needs further clarification. In addition, the frequency and type of IMT differed significantly among the included studies. The optimal training frequency and IMT type for specific populations

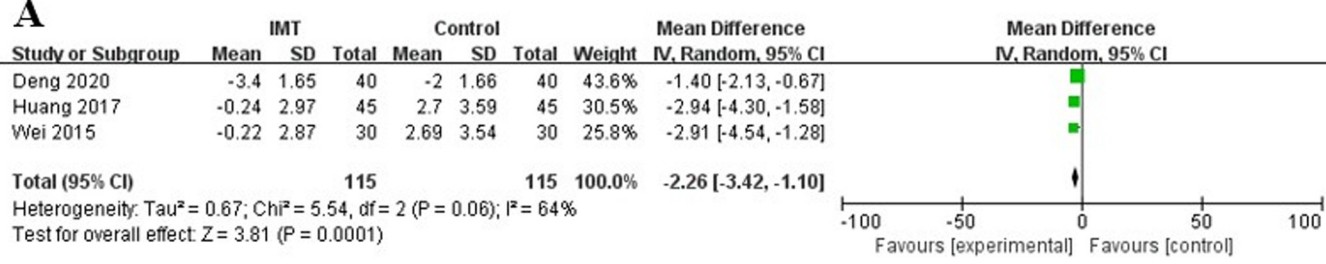

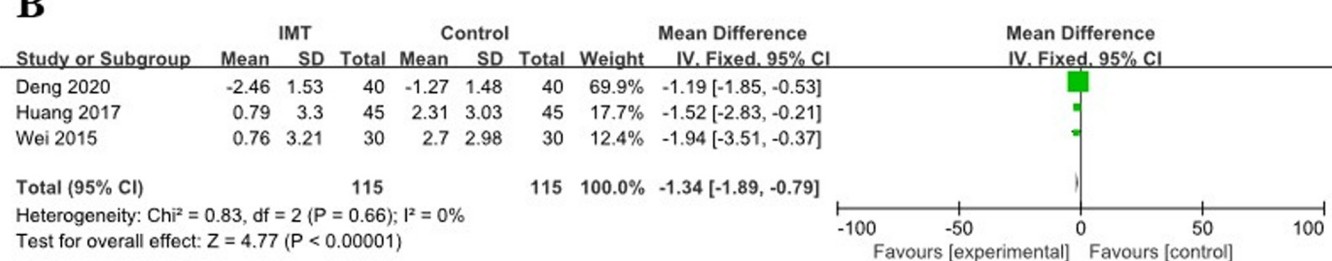

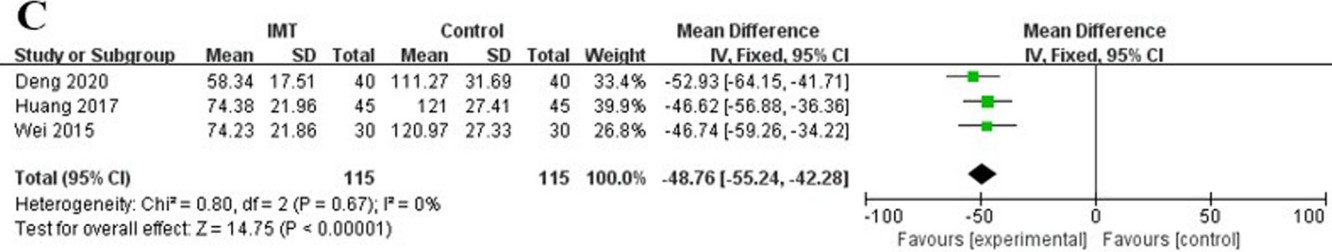

**Fig 5.** Forest plots for the effect of inspiratory muscle training on HADS anxiety score (A), HADS depression score (B) and NHP total score (C).

should be further identified. Furthermore, our results did not indicate that IMT reduced post-operative complications, which may be influenced by the small sample size. Therefore, large-sample studies are still needed to further clarify the correlation between IMT and the risk of postoperative complications.

Notably, our results indicate that IMT significantly improved several clinical outcomes, such as respiratory muscle function and lung function. However, with the introduction of the minimal clinically important difference (MCID) concept, whether IMT truly benefits patients clinically still requires more precise calculations. Additionally, the fact that patients in the control group included in the study did not receive any type of physical intervention may cause some bias in the results. Furthermore, the heterogeneity of multiple indicators was high, and we speculate that this may be related to factors such as small sample sizes in the included trials, different IMT intervention measures, and diverse populations included.

There are several limitations in this meta-analysis. First, the overall sample size of our meta-analysis was relatively small, which might cause bias. Second, some confounding factors existed in this meta-analysis, such as age, sex, type of surgery and IMT parameters. Third, we were unable to conduct a more detailed analysis based on the above parameters due to the lack of original data. Fourth, most of the included studies were from China. Fifth, in our meta-analysis, considerable heterogeneity was observed among some of the outcome measures, such as the MIP and 6MWT, which may to some extent affect the credibility of the results. Therefore,

more large-scale studies from other countries will still be needed in the future to further validate our findings and investigate the clinical value of IMT in different esophageal cancer populations.

## Conclusion

Therefore, our research results indicate that IMT might help improve postoperative respiratory function, exercise performance and quality of life in esophageal cancer patients. These findings provide some support for the widespread application of IMT in clinical practice. However, due to the limitations of this meta-analysis, more high-quality RCTs with larger sample sizes are needed to further determine the clinical value of IMT in surgical esophageal cancer patients.

## Author Contributions

**Conceptualization:** Jianhua Su, Pengming Yu.

**Data curation:** Jianhua Su, Wei Huang.

**Formal analysis:** Wei Huang.

**Methodology:** Jianhua Su.

**Software:** Wei Huang.

**Supervision:** Pengming Yu.

**Writing – original draft:** Jianhua Su.

**Writing – review & editing:** Wei Huang, Pengming Yu.

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
