## [Decision Letter · Decision Letter 0]

16 Feb 2024

PONE-D-23-43325Effect of inspiratory muscle training in esophageal cancer patients receiving esophagectomy: a meta-analysis of randomized controlled trials

PLOS ONE

Dear Dr. Yu,

Thank you for submitting your manuscript to PLOS ONE. After careful consideration, we feel that it has merit but does not fully meet PLOS ONE’s publication criteria as it currently stands. Therefore, we invite you to submit a revised version of the manuscript that addresses the points raised during the review process.

We look forward to receiving your revised manuscript.

Kind regards,

Mansueto Gomes Neto, Ph.D

Academic Editor

PLOS ONE

Journal Requirements:

Reviewers' comments:

Reviewer's Responses to Questions

**Comments to the Author**

1. Is the manuscript technically sound, and do the data support the conclusions?

Reviewer #1: Partly

Reviewer #2: Yes

Reviewer #3: No

2. Has the statistical analysis been performed appropriately and rigorously? 

Reviewer #1: Yes

Reviewer #2: Yes

Reviewer #3: Yes

3. Have the authors made all data underlying the findings in their manuscript fully available?

Reviewer #1: No

Reviewer #2: No

Reviewer #3: Yes

4. Is the manuscript presented in an intelligible fashion and written in standard English?

Reviewer #1: No

Reviewer #2: No

Reviewer #3: Yes

5. Review Comments to the Author

Reviewer #1: The manuscript needs language revision by a native English-speaking editor. Some section read well, but others need attention.

Introduction:

This should include the effect of IMT on esophageal cancer patients. What is the gap in the study? In addition, using IMT is not common in esophageal cancer, why did you decide to review it? what was the importance of the study?

Please provide the questions address by the review.

Materials and methods:

Please define IMT (e.g., type), and other outcomes (i.e., postoperative complications; how did you report or measure it).

Results:

Table 1; please provide types of IMT in Table 1.

Discussion:

Page 7, Line 17: you stated that "Cordeiro ALL included seven studies ..." Please check this sentence? that might be a typing error.

Page 8, line 7: you stated that " ... and the study by Guinan et al. showed no significant value of IMT.." Please provide the reference.

line 12-13: you stated that "Secondly, the clinical outcome measures analyzed in the three foreign studies were generally limited." Please provide further details.

Line 19-26; this might also be discussed in terms of frequency, type of IMT, and duration of IMT.

Overall, the conclusion is certainly vague. In addition, the discussion part need to revised.

Reviewer #2: Dear authors,

I had the opportunity to review the paper entitled “Effect of inspiratory muscle training in esophageal cancer patients receiving esophagectomy: a meta-analysis of randomized controlled trials”

After reading the manuscript I have some concerns that should be resolved:

-Proofreading is crucial to catch typographical errors and ensure the professionalism of the manuscript.

Abstract:

The abstract provides a brief overview of the study.

Introduction:

-The sentence "Although minimally invasive approaches have become increasingly prevalent in the surgical treatment of esophageal cancer in recent decades, surgery still results in significant trauma" is somewhat awkward. Consider rephrasing for better clarity, such as "Despite the increasing prevalence of minimally invasive approaches in recent decades, esophageal cancer surgery continues to result in significant trauma."

-In the sentence "The conventional respiratory training methods currently used in clinical practice 16 mainly include diaphragmatic breathing, blowing up balloons, and lip puckering breathing," it might be clearer to say "commonly" instead of "mainly" for smoother phrasing.

-The mention of "Several relevant randomized controlled trials (RCTs) explored the clinical effect of IMT in esophageal cancer patients receiving esophagectomy, but they have shown significant differences in their results" could be more specific. What kind of differences are being referred to? Clarity here will better prepare the reader for the objective of the meta-analysis.

Materials and Methods:

-The inclusion of specific databases in the literature search is comprehensive. However, the abbreviation "CNKI" is used without explanation. Consider providing the full name on the first mention, e.g., "China National Knowledge Infrastructure (CNKI)."

-In the inclusion criteria, the phrase "patients received the esophagectomy and were diagnoses with primary esophageal cancer" contains a grammatical error ("diagnoses" should be "diagnosed"). Consider revising for accuracy: "patients received esophagectomy and were diagnosed with primary esophageal cancer."

-Consider providing a language preference for the inclusion criteria.

-The exclusion criteria are clear, but you might want to elaborate on what is meant by "low-quality studies" described below, or provide specific criteria for defining low quality.

-The information extracted from each study is comprehensive. However, the sentence structure could be improved for clarity. For instance, "and detailed data about observation endpoints" might benefit from clarification or rephrasing.

-Consider using Cochrane risk of bias assessment (RoB 2.0, Cochrane, London, UK) for risk of bias assessment.

-The clarification on how discrepancies were resolved during the literature search, selection, data collection, and quality assessment is good practice. It demonstrates transparency and reliability in the review process.

Result:

- More information about included the number of articles retrieved initially and the number included after screening is necessary.

-Did you use the Preferred Reporting Items for Systematic Reviews and Meta-Analysis (PRISMA 2020) flow diagram?

-In the sentence "Most include studies were from China," there's a minor grammatical error. It should be "Most included studies were from China."

-Consider to give more information about interventions.

-In the sentence "Similarly, IMT played a role in 1 improving the pulmonary function...," there's a typographical error. It should be "Similarly, IMT played a role in improving pulmonary function."

-In the sentence "In this meta-analysis, the 6MWT and Borg index were applied to assess the exercise performance status," it would be beneficial to briefly explain why these specific metrics were chosen.

-The organization of the section on postoperative complications is clear, but you could add a brief statement about why assessing these complications is relevant in the context of the study.

-Provide a brief information about Heterogeneity.

Discussion:

-In the sentence "Therefore, perioperative IMT could significantly improve clinical outcomes...," consider specifying the particular clinical outcomes that are significantly improved. It adds clarity and helps the reader understand the impact of IMT.

-The comparison with previous meta-analyses is valuable, providing context and support for the current study. However, it would be beneficial to briefly discuss any differences in methodologies, populations studied, or outcomes assessed between your study and the referenced meta-analyses.

-Consider providing a brief explanation for the cited meta-analyses' methodologies and populations studied, especially for readers who may not be familiar with these studies.

-In the sentence "Thus, IMT may also have significant clinical value in esophageal cancer surgery...," consider specifying the aspects of esophageal cancer surgery where IMT demonstrates clinical value.

-The acknowledgment of differences between domestic and foreign studies adds depth to the discussion. However, it would be helpful to provide a bit more detail on the potential reasons for these differences, perhaps exploring the variation in healthcare practices, patient characteristics, or study methodologies.

-The speculation about the combination of preoperative and postoperative training yielding better results is interesting. However, you might want to phrase it as a hypothesis and suggest it as a direction for future research rather than stating it as a conclusion.

-The discussion about the limitations is essential. However, it's recommended to provide potential solutions or suggestions for mitigating these limitations, such as advocating for larger sample sizes in future studies.

-The point about the optimal timing and training regimen is crucial. Consider expanding on the existing literature or theories that discuss these aspects to provide more context and rationale.

-In the conclusion, you state, "and might be widely applied in clinics." Consider reinforcing this statement with a brief summary of the key findings that support the widespread application of IMT in clinical settings.

-The conclusion could benefit from a more explicit restatement of the study's main contribution or novelty, emphasizing why these findings are important for the field.

Reviewer #3: Dear authors,

Thank you for submitting the manuscript "Effect of inspiratory muscle training in esophageal cancer patients receiving esophagectomy: a meta-analysis of randomized controlled trials".

What are the Major Strengths of the Paper: The major strengths of the paper is summarize studies using inspiratory muscle training.

Major Recommendations:

METHODS

- Informations about Study Selection and research strategy are necessary.

- Additionally, how many researchers participated in data extraction and study selection?

- Is important highlighted: No record of the research protocol was considered. Has the current research protocol been registered on the Prospero platform?.

-

RESULTS

- Despite the statistical difference observed, it is worth highlighting that: 1. most results appear to be below the MCID; 2. there is a great imprecision for the most encouraging result, which would be that of the 6-minute walk test, thus reducing confidence in the finding; 3. heterogeneity is high for most results, thus reducing confidence in the finding.

DISCUSSION

- The discussion needs to critically analyze the results reported: MCID (how much the statistical difference is reflected in the real clinical difference) grade (how much these included studies give me strength to trust this statistical difference), diversity of training protocols and the fact that the control group was not subjected to any type of physical intervention.

- Based on what was observed, it is relevant to discuss what the authors believe may have contributed to the high heterogeneity of some findings.

CONCLUSION

- Another important point: the conclusion goes beyond what the results can say. Considering that the MCID for none of the outcomes was mentioned and the level of evidence was not assessed by GRADE

-Finally, I highlight that given the weaknesses on the subject (which can be seen mainly by small effect magnitudes, high imprecision and great heterogeneity), a more considered conclusion about the effects of inspiratory muscle training is appropriate

6. PLOS authors have the option to publish the peer review history of their article (what does this mean?). If published, this will include your full peer review and any attached files.

Reviewer #1: No

Reviewer #2: No

Reviewer #3: No

---

## [Author Response · Author response to Decision Letter 0]

26 Mar 2024

Response to Reviewer #1: 

Question 1: The manuscript needs language revision by a native English-speaking editor. Some section read well, but others need attention.

Answer 1: Thanks for your comment. This manuscript has been edited by AJE service with the verification code E393-49BF-F3F3-03C8-F6F0.

Question 2: Introduction: This should include the effect of IMT on esophageal cancer patients. What is the gap in the study? In addition, using IMT is not common in esophageal cancer, why did you decide to review it? what was the importance of the study?

Please provide the questions address by the review.

Answer 2: Thanks for your valuable question. We added the introduction about this issue in the manuscript as follows “Esophageal cancer surgery may involve the removal of part of the esophagus or adjacent organs, affecting the tissues within the chest cavity. This can impact the patient's respiratory function, including lung capacity and respiratory muscle strength. IMT can help patients regain respiratory function and increase lung capacity and respiratory muscle strength [13, 14]. Following esophageal cancer surgery, patients may experience respiratory complications such as lung infections, pneumonia, or atelectasis. These complications can prolong recovery time and increase patient discomfort. Inspiratory muscle training can help prevent the occurrence of these respiratory complications, improving the success rate of recovery [15]. Esophageal cancer surgery is a traumatic procedure that can lead to physical weakness and muscle atrophy in patients. IMT can assist patients in increasing muscle strength and endurance, accelerating the recovery process and reducing postoperative complications [16]. Besides, surgery is a crucial treatment method for esophageal cancer, but it may also have certain impacts on patients' lives, such as restricted activity, pain, and discomfort. Through IMT, patients can enhance their physical fitness, improve their quality of life, and better adapt to life after surgery. Therefore, theoretically, IMT may also bring significant clinical value to esophageal cancer surgery patients, but currently there is a lack of robust strong evidence.”

Question 3: Materials and methods: Please define IMT (e.g., type), and other outcomes (i.e., postoperative complications; how did you report or measure it).

Answer 3: Thanks for your comments. We have modified the inclusion criteria. The type of IMT has been defined. However, in some included studies, the criteria for postoperative complications were not described. Therefore, in this meta-analysis, we were unable to define standard categorizations. Actually, this is a common issue in meta-analyses (secondary research).

Question 4: Results: Table 1; please provide types of IMT in Table 1.

Answer 4: After re-reviewing included studies, all patients received IMT through the respiratory training device. This information has been added in the table 1.

Question 5: Discussion: Page 7, Line 17: you stated that "Cordeiro ALL included seven studies ..." Please check this sentence? that might be a typing error.

Answer 5: This sentence has been modified and changed to “Cordeiro et al.”.

Question 6: Page 8, line 7: you stated that " ... and the study by Guinan et al. showed no significant value of IMT.." Please provide the reference.

Answer 6: Sorry for this mistake. We have added the corresponding reference.

Question 7: line 12-13: you stated that "Secondly, the clinical outcome measures analyzed in the three foreign studies were generally limited." Please provide further details.

Answer 7: We have added detailed explanation for this sentence.” In detail, the studies by Van Adrichem and Valkenet et al. only explored the correlation between preoperative IMT and postoperative complications [24, 28].”

Question 8: Line 19-26; this might also be discussed in terms of frequency, type of IMT, and duration of IMT.

Answer 8: Thanks for your valuable suggestion. We have added this discussion in the manuscript as follows:” The optimal training regimen, including training time and duration, for different populations also needs further clarification. Besides, the frequency and type of IMT in included studies significantly differed. The optimal training frequency and IMT type for specific populations should be further identified.”

Question 9: Overall, the conclusion is certainly vague. In addition, the discussion part need to revised.

Answer 9: Thanks for your question. We have revised the conclusion and discussion part.

Response to Reviewer #2:

After reading the manuscript I have some concerns that should be resolved:-Proofreading is crucial to catch typographical errors and ensure the professionalism of the manuscript. Abstract: The abstract provides a brief overview of the study.

Question 1: Introduction:

-The sentence "Although minimally invasive approaches have become increasingly prevalent in the surgical treatment of esophageal cancer in recent decades, surgery still results in significant trauma" is somewhat awkward. Consider rephrasing for better clarity, such as "Despite the increasing prevalence of minimally invasive approaches in recent decades, esophageal cancer surgery continues to result in significant trauma."

Answer 1: Thanks for your valuable suggestion. This sentence has been modified as suggested.

Question 2: In the sentence "The conventional respiratory training methods currently used in clinical practice 16 mainly include diaphragmatic breathing, blowing up balloons, and lip puckering breathing," it might be clearer to say "commonly" instead of "mainly" for smoother phrasing.

Answer 2: The term “mainly” has been changed to “commonly”.

Question 3: The mention of "Several relevant randomized controlled trials (RCTs) explored the clinical effect of IMT in esophageal cancer patients receiving esophagectomy, but they have shown significant differences in their results" could be more specific. What kind of differences are being referred to? Clarity here will better prepare the reader for the objective of the meta-analysis.

Answer 3: We have added detailed description here as follows:” Several relevant randomized controlled trials (RCTs) explored the clinical effect of IMT in esophageal cancer patients receiving esophagectomy, but they have shown significant differences such as the impact of IMT on the respiratory muscle function and pulmonary function.”

Question 4: Materials and Methods: The inclusion of specific databases in the literature search is comprehensive. However, the abbreviation "CNKI" is used without explanation. Consider providing the full name on the first mention, e.g., "China National Knowledge Infrastructure (CNKI)."

Answer 4: Thanks for your comment. We have added the full name of CNKI.

Question 5: In the inclusion criteria, the phrase "patients received the esophagectomy and were diagnoses with primary esophageal cancer" contains a grammatical error ("diagnoses" should be "diagnosed"). Consider revising for accuracy: "patients received esophagectomy and were diagnosed with primary esophageal cancer."

Answer 5: The term “diagnoses” has been changed to “diagnosed”.

Question 6: Consider providing a language preference for the inclusion criteria.

Answer 6: We have added the language limitation as one of the inclusion criteria.

Question 7: The exclusion criteria are clear, but you might want to elaborate on what is meant by "low-quality studies" described below, or provide specific criteria for defining low quality.

Answer 7: We have added the description about “low-quality studies” as follows:” low-quality studies with the PEDro score of 3 or lower [19, 20].”

Question 8: The information extracted from each study is comprehensive. However, the sentence structure could be improved for clarity. For instance, "and detailed data about observation endpoints" might benefit from clarification or rephrasing.

Answer 8: Thanks for your valuable question. We have modified this sentence as follows: “ Following information was extracted from each included studies: the name of first author, publication year, country, number of participants, type of surgery, type of IMT, age, intervention time, initial training pressure, training time, sessions and duration of IMT, control care, endpoints, information about PEDro scale and detailed data about observation endpoints including the mean differences (MDs) and odds ratios (ORs) with 95% confidence intervals (CIs).”

Question 9: Consider using Cochrane risk of bias assessment (RoB 2.0, Cochrane, London, UK) for risk of bias assessment.

Answer 9: Thanks for your question. Cochrane risk of bias assessment is a commonly used tool for the meta-analyses based on RCTs. Actually, PEDro scale is a novel tool primarily used for assessing risk of bias in meta-analyses of rehabilitation studies. In fact, the basic principles and evaluation criteria of these two tools are quite similar, and we have also provided specific scoring information in Table 2.

Question 10: More information about included the number of articles retrieved initially and the number included after screening is necessary.

Answer 10: We have modified this part as follows: “ As shown in figure 1, 77 records were identified from several databases initially and 25 duplicated records were removed. After reviewing the titles and abstracts, 41 publications were excluded. Then after reviewing the full texts and excluding three studies, eight RCTs were eventually included in this meta-analysis”.

Question 11: Did you use the Preferred Reporting Items for Systematic Reviews and Meta-Analysis (PRISMA 2020) flow diagram?

Answer 11: As we mentioned in the “Materials and methods”, “The current meta-analysis was performed according to the Preferred Reporting Items for Systematic Review and Meta-Analyses 2020”. Besides, figure 1 presented the prima flow diagram of this meta-analysis.

Question 12: In the sentence "Most include studies were from China," there's a minor grammatical error. It should be "Most included studies were from China."

Answer 12: Sorry for this mistake. It has been corrected.

Question 13: Consider to give more information about interventions.

Answer 13: Thanks for your valuable question. We have added the information about interventions in the manuscript as follows:” Preoperative IMT was involved in most patients. Besides, 60% of initial training pressure and seven sessions a week were applied in most included studies. The durations of IMT ranged from 10 days to 4 weeks.” 

Question 14: In the sentence "Similarly, IMT played a role in 1 improving the pulmonary function...," there's a typographical error. It should be "Similarly, IMT played a role in improving pulmonary function."

Answer 14: This mistake has been corrected.

Question 15: In the sentence "In this meta-analysis, the 6MWT and Borg index were applied to assess the exercise performance status," it would be beneficial to briefly explain why these specific metrics were chosen.

Answer 15: We have added the brief description about 6MWT and Borg index. “Based on previous research, the 6MWT and Borg scale are widely utilized for assessing exercise performance status among patients receiving rehabilitation training, demonstrating good applicability and reliability. Therefore, in this meta-analysis, the 6MWT and Borg index were applied to assess the exercise performance status.”

Question 16: The organization of the section on postoperative complications is clear, but you could add a brief statement about why assessing these complications is relevant in the context of the study.

Answer 16: Actually, we analyzed all complications based on available data from included studies. This issue has been stated as follows: “We assessed the effect of IMT on postoperative complication based on available data provided in included studies.”

Question 17: Provide a brief information about Heterogeneity.

Answer 17: The information about heterogeneity has been added.

Question 18: Therefore, perioperative IMT could significantly improve clinical outcomes...," consider specifying the particular clinical outcomes that are significantly improved. It adds clarity and helps the reader understand the impact of IMT.

Answer 18: This sentence has been modified as follows: “Therefore, perioperative IMT could significantly improve clinical outcomes such as the respiratory muscle function and pulmonary function of esophageal cancer patients and facilitate patient recovery after surgery, which shows high clinical application potentials.”

Question 19: The comparison with previous meta-analyses is valuable, providing context and support for the current study. However, it would be beneficial to briefly discuss any differences in methodologies, populations studied, or outcomes assessed between your study and the referenced meta-analyses.

Answer 19: Thanks for your valuable suggestion. We have added the brief description as follows: “However, these meta-analyses included patients undergoing different surgical procedures and varied outcome measures, thus failing to provide conclusive evidence on whether esophageal cancer patients should receive IMT.”

Question 20: Consider providing a brief explanation for the cited meta-analyses' methodologies and populations studied, especially for readers who may not be familiar with these studies.

Answer 20: Thanks for your valuable suggestion. We have added more details about the methodologies and populations studied of cited meta-analyses.

Question 21: In the sentence "Thus, IMT may also have significant clinical value in esophageal cancer surgery...," consider specifying the aspects of esophageal cancer surgery where IMT demonstrates clinical value.

Answer 21: We have modified this sentence as follows: “Thus, IMT may also have significant clinical value among patients receiving esophageal cancer surgery in improving postoperative respiratory function and reducing the risk of postoperative complications.”

Question 22: The acknowledgment of differences between domestic and foreign studies adds depth to the discussion. However, it would be helpful to provide a bit more detail on the potential reasons for these differences, perhaps exploring the variation in healthcare practices, patient characteristics, or study methodologies.

Answer 22: We have added the description about this issue. “For example, the pathological types of esophageal cancer and the prevalence of minimally invasive surgery vary between different countries.”

Question 23: The speculation about the combination of preoperative and postoperative training yielding better results is interesting. However, you might want to phrase it as a hypothesis and suggest it as a direction for future research rather than stating it as a conclusion.

Answer 23: Thanks for your question. We have stated this issue as request. “compared to preoperative training alone, the combination of preoperative and postoperative training may yield better results, but this needs further confirmation by more high-quality RCTs.”

Question 24: The discussion about the limitations is essential. However, it's recommended to provide potential solutions or suggestions for mitigating these limitations, such as advocating for larger sample sizes in future studies.

Answer 24: Thanks for your question. We have added the description as follows: “Therefore, more large-scale studies from other countries will still be needed in the future to further validate our findings and investigate the clinical value of IMT in different esophageal cancer populations.”

Question 25: In the conclusion, you state, "and might be widely applied in clinics." Consider reinforcing this statement with a brief summary of the key findings that support the widespread application of IMT in clinical settings.

Answer 25: We have modified the conclusion as follows “Therefore, our research results indicate that IMT might help improve postoperative respiratory function, exercise performance and quality of life in esophageal cancer patients. These findings provide some support for the widespread application of IMT in clinical practice. However, due to the limitations exist in this meta-analysis, more high-quality RCTs with bigger sample sizes are needed

---

## [Decision Letter · Decision Letter 1]

1 Jul 2024

Effect of inspiratory muscle training in esophageal cancer patients receiving esophagectomy: A meta-analysis of randomized controlled trials

PONE-D-23-43325R1

Dear Dr. Yu,

We’re pleased to inform you that your manuscript has been judged scientifically suitable for publication and will be formally accepted for publication once it meets all outstanding technical requirements.

Kind regards,

Fatma Abdelfattah Hegazy, Ph.D.

Academic Editor

PLOS ONE

Additional Editor Comments (optional):

Reviewers' comments:

Reviewer's Responses to Questions

**Comments to the Author**

1. If the authors have adequately addressed your comments raised in a previous round of review and you feel that this manuscript is now acceptable for publication, you may indicate that here to bypass the “Comments to the Author” section, enter your conflict of interest statement in the “Confidential to Editor” section, and submit your "Accept" recommendation.

Reviewer #1: All comments have been addressed

Reviewer #2: All comments have been addressed

2. Is the manuscript technically sound, and do the data support the conclusions?

Reviewer #1: Yes

Reviewer #2: Yes

3. Has the statistical analysis been performed appropriately and rigorously? 

Reviewer #1: Yes

Reviewer #2: Yes

4. Have the authors made all data underlying the findings in their manuscript fully available?

Reviewer #1: Yes

Reviewer #2: Yes

5. Is the manuscript presented in an intelligible fashion and written in standard English?

Reviewer #1: Yes

Reviewer #2: Yes

6. Review Comments to the Author

Reviewer #1: (No Response)

Reviewer #2: The authors have responded to my comments before. No further questions to the authors about manuscript.

7. PLOS authors have the option to publish the peer review history of their article (what does this mean?). If published, this will include your full peer review and any attached files.

Reviewer #1: No

Reviewer #2: No

---

## [Editor Report · Acceptance letter]

4 Jul 2024

PONE-D-23-43325R1 

PLOS ONE

Dear Dr. Yu, 

I'm pleased to inform you that your manuscript has been deemed suitable for publication in PLOS ONE. Congratulations! Your manuscript is now being handed over to our production team.

Kind regards, 

on behalf of

Prof. Fatma Abdelfattah Hegazy 

Academic Editor

PLOS ONE